

# Peat-DBase v.1: A Compiled Database of Global Peat Depth Measurements

Jade Skye[1,2], Joe R. Melton[1,2], Colin Goldblatt[2], Louis Saumier[2], Angela Gallego-Sala[3],
Michelle Garneau[4], R. Scott Winton[5], Erick B. Bahati[6,7], Juan C. Benavides[8], Lee Fedorchuk[9],
Gérard Imani[6], Carol Kagaba[10], Frank Kansiime[11], Mariusz Lamentowicz[12], Michel Mbasi[13],
Daria Wochal[12], Sambor Czerwiński[14], Jacek Landowski[15], Joanna Landowska[16], Vincent Maire[17],
Minna M. Väliranta[18], Matthew Warren[19], Lydia E. S. Cole[20], Marissa A. Davies[21], Erik A. Lilleskov[22],
Jingjing Sun[23], and Yuwan Wang[3,24]

[1]Climate Research Division, Environment and Climate Change Canada, Victoria BC, Canada
[2]School of Earth and Ocean Sciences, University of Victoria, Canada
[3] Department of Geography, Faculty of Economy, Science and the Environment, University of Exeter, Exeter, UK
[4] Department of geography and Geotop research center. Université du Québec à Montréal, Canada
[5]Environmental Studies Department, University of California Santa Cruz, California, United States
[6]Department of Biology, Faculty of Sciences, Université Officielle de Bukavu, Bukavu, DR Congo
[7]Centre de recherche en écologie et gestion des écosystèmes terrestres, Université Officielle de Bukavu
[8]Pontificia Universidad Javeriana, Bogotá Colombia
[9]Peatlands Program, Forestry and Peatlands Branch, Goverment of Manitoba, Canada
[10]Ministry of Water and Environment, Uganda
[11]Department of Environmental Management, University of Makerere, Kampala, Uganda
[12]Climate Change Ecology Research Unit, Faculty of Geographical and Geological Sciences, Adam Mickiewicz University in Poznań, Poland
[13]Laboratory of Ecology and Sustainable Forest Management, University of Kisangani
[14]Department of Geomorphology and Quarternary Geology, Faculty of Oceanography and Geography, University of Gdańsk, Gdańsk 81-378, Poland
[15]Szkoła Podstawowa nr 44 im. Obrońców Wybrzeża w Gdyni, Poland
[16]XIV Liceum Ogólnokształcące z Oddziałami Dwujęzycznymi im. Mikołaja Kopernika w Gdyni, Gdynia, Poland
[17]Département des sciences de l'environnement, Université du Québec à Trois-Rivières, Canada
[18]Environmental Change Research Unit, Ecosystems and Environment Research Programme, Faculty of Biological and Environmental Sciences, University of Helsinki, Finland
[19]Food and Agriculture Organization of the United Nations, Rome, Italy
[20]School of Geography and Sustainable Development, University of St. Andrews, Scotland
[21]University of Waterloo, Canada
[22]Forest Service, U.S. Department of Agriculture
[23]State Key Laboratory of Black Soils Conservation and Utilization, Northeast Institute of Geography and Agroecology, Chinese Academy of Sciences, China
[24]School of the Environment, University of Queensland, Brisbane, Australia

**Correspondence:** Joe R. Melton (joe.melton@ec.gc.ca)

**Abstract.** Peatlands are globally important carbon stores that face increasing threats from human activities and climate change impacts. Comprehensive peatland data are essential for understanding ecosystem responses to these stressors and mapping their past and current characteristics. Current peatland datasets remain limited due to poor representation in global soil mapping



initiatives and the absence of a recognized, coordinated central repository for peat depth data. Existing compilations often
contain errors, duplicates, and outdated observations, requiring researchers to repeatedly gather and harmonize data on a
study-by-study basis. To address these challenges, we present Peat-DBase version 1.0—a harmonized, quality-controlled global
compilation of basal peat depth measurements.

Version 1.0 of Peat-DBase comprises 204 902 peat depth measurements from 29 sources spanning 54.933°S to 82.217°N,
with a significant proportion of measurements in Atlantic Canada and Scotland due to the inclusion of two particularly large
datasets focused on those regions. We supplement the peat study measurements with 94 615 non-peat soil measurements
to ensure comprehensive coverage consistent with the relatively low spatial coverage of peatlands globally. Despite the un-
even distribution of peat depth measurements, Peat-DBase contains reasonable coverage of the major global peatland com-
plexes in temperate and boreal North America and Europe, portions of Russia, the Amazon and Congo basins, and the Malay
Archipelago, though gaps remain in the lower Amazon Basin, Eastern Indonesia, and Eastern Russia. From the current data,
peat depths average 144 cm, although this is influenced by a predominance of measurements in the North Atlantic regions.
Peat-DBase's deepest measurement is 3 527 cm.

While sampling biases and measurement uncertainties exist, Peat-DBase provides an essential foundation for global peat-
land research. Peat-DBase is under active development and future versions will incorporate additional datasets, information
on current peatland status, and improved positional uncertainty quantification. Peat-DBase eliminates the need for overlap-
ping data compilation efforts while identifying critical observational gaps for future research. Peat-DBase is available at
https://doi.org/10.5281/zenodo.15530645

# 1  Introduction

Peatlands are important carbon (C) stores, comprising roughly one-third of global soil C despite only covering about 3% of
Earth's land surface (Joosten and Clarke, 2002; Jackson et al., 2017; Xu et al., 2018; Melton et al., 2022). They also contain
about 10% of the world's fresh surface water and support unique biodiversity (Joosten and Clarke, 2002; Page and Baird, 2016).
Their persistently saturated conditions inhibit decomposition, enabling peat accumulation over centuries to millennia (Koster
and Favier, 2005; Page and Baird, 2016; Joosten and Clarke, 2002). However, these ecosystems are sensitive to anthropogenic
activity and climate change impacts (Loisel et al., 2021). Drainage for agriculture, forestry, and other land uses can alter
peatland soil structure and function, lower water tables, promote aerobic decomposition, and dramatically increase carbon
emissions (Fluet-Chouinard et al., 2023; Page and Baird, 2016; Warren et al., 2017; Koster and Favier, 2005; Li et al., 2018).





Climate change intensifies these threats through rising temperatures and altered hydrologic regimes, with extreme weather events often leading to increased wildfire risk (Canadell et al., 2021; Helbig et al., 2020).

Comprehensive peatland data – including extent (Xu et al., 2018; Melton et al., 2022; Gumbricht et al., 2017), carbon content (Gorham, 1991; Page et al., 2011), protection status (Austin et al., 2025), land-use, land cover, and degree of degradation (Fluet-Chouinard et al., 2023) – are urgently needed to understand ecosystem responses to current environmental and anthro-
pogenic stressors and project future ecosystem dynamics. As peatlands are increasingly incorporated into land surface models (Chadburn et al., 2022; Bechtold et al., 2019b; Wu et al., 2016) for eventual integration into Earth system models, as well as data-driven mapping applications (e.g., Minasny et al., 2019; Melton et al., 2022), robust peatland datasets are essential for model initialization and evaluation. However, peatland data remain scarce due to under-representation in global soil mapping initiatives (Krankina et al., 2008; Minasny et al., 2019) and the characteristically wet, and often remote, locations of peatlands,
which create challenging conditions for field data collection (Minasny et al., 2019; Rudiyanto et al., 2016). Peat depth mapping is particularly difficult as it requires labour-intensive field surveys or proximal remote sensing techniques (Minasny et al., 2019; Jowsey, 1966).

To address these needs, we present Peat-DBase version 1.0—a harmonized, quality-controlled global compilation of basal peat depth measurements. The database also includes mineral soil core data (Batjes et al., 2020b) to provide comprehensive
coverage of peat-free areas. The rest of our paper is structured as follows. First, in Sect. 2, we explain the intentions of Peat-DBase and why it is needed now to advance global peatland science. Next, in Sect. 3, we describe the present state of Peat-DBase, the protocols for the acquisition, formatting, and processing for the peat study data; mineral soil data is covered in Appendix Sect. A. Then, in Sect. 4, we provide an analysis and discussion of the resulting database, including its limitations and our future plans for Peat-DBase. Finally, in Sect. 5 and 6, we discuss the conclusions and data availability, respectively.

## 2   Motivation: Why Peat-DBase and why now?

While Peat-DBase v1.0 focuses on peat depth, our vision is to establish it as a comprehensive global repository for peatland carbon information. Our intention is to complement similar community-led initiatives like the Soil Respiration Database (SRDB; Jian et al., 2020), World Soil Information Service (WoSIS; Batjes et al., 2020a), and the International Soil Radiocarbon Database (ISRaD Lawrence et al., 2020), but with a focus on peatlands and their carbon stocks. Current peatland compilations
assembled for individual studies (e.g. Gorham et al., 2012; Treat et al., 2019; Hugelius et al., 2020) often contain errors, duplicates, outdated observations, among other issues. By creating a curated, open-source database, we eliminate the need for researchers to repeatedly gather, clean and harmonize data independently. Peat-DBase also does not duplicate the work of other peatland databases, such as "The Global Peatland Database" (https://greifswaldmoor.de/global-peatland-database-en.html), which focuses on peatland extent and drainage status rather than depth measurements and does not permit access to the datasets
behind the released map products; or "PeatData Hub" (https://peatdatahub.net/data-packages/; Xu et al., 2018), which is also focused on peatland extent and water table depths.





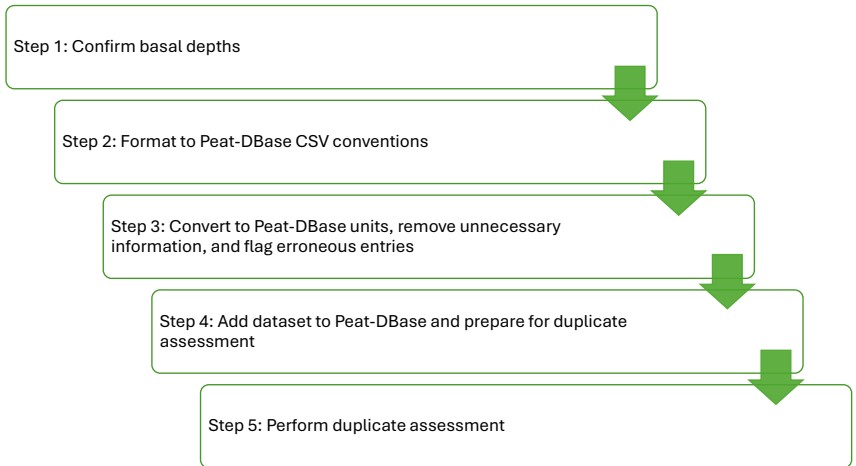

**Figure 1.** Workflow to ingest peat-focused datasets into Peat-DBase

Peat-DBase is particularly timely for several reasons. First, advancing peatland representation in land surface models (Wu et al., 2016; Bechtold et al., 2019a; Apers et al., 2022) requires accurate, extensive data for initialization and evaluation, especially as these models are integrated into Earth system models. Second, the proliferation of artificial intelligence (AI) techniques in geosciences (Wadoux et al., 2020) has enabled data-driven mapping of peatland extent (e.g., Minasny et al., 2019; Melton et al., 2022), carbon stocks (Hugelius et al., 2020; Widyastuti et al., 2025), and depth (Skye, 2025; Widyastuti et al., 2025). However, AI model outputs depend critically on training data quality and quantity (Aroyo et al., 2022; Roscher et al., 2024). Since these approaches cannot reliably extrapolate beyond their training domains (Xu et al., 2020; Hateffard et al., 2024), providing spatially extensive, high-quality datasets is essential. Third, despite facing intense pressures from human activities and climate change, peatlands remain poorly mapped, hindering conservation and restoration efforts (Austin et al., 2025). Fourth, as part of the Paris Agreement, 20 countries have already listed peatland or wetland soil organic carbon mitigation targets in their Nationally Determined Contributions (NDCs) (Wiese et al., 2021). However, many other countries did not include peatland or wetland soils in their NDCs, citing, among other reasons, the difficulty in accurately quantifying and monitoring change in these stocks (Wiese et al., 2021). Finally, global databases like Peat-DBase can identify critical observational gaps, motivating targeted research and data collection efforts.

## 3   Methods: Peat Data Compilation and Processing

The workflow to compile and process new peat depth datasets is shown in Figure 1. The individual steps will be described in the following sections.



## 3.1 Data Acquisition

Peat study data were accepted into Peat-DBase provided the measurements were taken down to the basal depth indicated by mineral soil or bedrock; beyond this requirement, any coring or sampling method was allowed that was able to accurately determine the basal depth[1]. In permafrost regions, Cold Regions Research and Engineering Laboratory corers (Brockett and Lawson, 1985) were often used. In non-permafrost sites, Russian-type corers (Jowsey, 1966), Box corers (Shotyk and Noernberg, 2020; Fenton, 1980), and Jeglum corers (Jeglum et al., 1991) were the primary tools. In cases where pole probing was 90 conducted as part of a coring transect, the probing measurements were also included. Probing, involves the use of metal poles which are inserted into the peat until they meet a non-peat layer and cannot go any further (e.g. Oakfield probes; Magnan et al., 2024; Householder et al., 2012; Crezee et al., 2022).

The sampling protocol varied depending on the goals of the researchers. In some cases, transects of varying lengths were chosen with measurements taken at consistent intervals across the transect (e.g. Crezee et al., 2022; Kelly et al., 2020; Winton 95 et al., 2025). In other instances, unique core sites were chosen across single or multiple peatlands (e.g. Cole et al., 2015; Davies et al., 2023b, a; Silvestri et al., 2019a). Some peat study datasets also included some cores taken in peat-free soils as a result of their sampling procedures (e.g. Crezee et al., 2022; Keys and Henderson, 1987; Thibault, 1992). Other data sources were extensive compilations of numerous field campaigns with different goals and objectives yielding comparably dense measurement coverage (e.g. Keys and Henderson, 1987; Thibault, 1992; Scottish Government, 2025).

Peat depth data available from government or non-governmental agencies were the largest sources of data in terms of number of measurements. The two principal datasets were made available for download online (e.g. NatureScot, n=174 159; accessed on May 26, 2025) or associated with publications where the data was available upon request (e.g. the Government of New Brunswick, n = 20 505; Keys and Henderson, 1987; Thibault, 1992).

Generally, the largest peat depth datasets derived from the scientific literature were compilations of other datasets. Such 105 compiled datasets were typically developed for modelling purposes (e.g. Hugelius et al., 2020; Treat et al., 2017, 2019). As these compiled datasets can often have convoluted histories – frequently incorporating other compilations – they were added to Peat-DBase under the single citation of the compiling authors. Additional fields were used to track any information the compiling authors provided regarding their data sources (see Sect. 3.2 and Table 2).

A total of 204 902 peat depth measurements came from 29 sources (all totals are after error/duplicate assessment and listed 110 in Table 1). Of these, four sources are previously unpublished representing 2 352 measurements. Some publications were excluded from Peat-DBase because their underlying data were not readily accessible, lacked usable data files, or we could not confirm from the publication that the measurements represented basal peat depth. All sources of peat depth measurements used in Peat-DBase are listed in Table 1.

---

[1]Future versions of Peat-DBase will allow peat data that does not reach the basal depth through the use of appropriate flags indicating that fact.





**Table 1.** The sources of peat study measurements in Peat-DBase v. 1. The final number indicates the number of measurements (n) retained after error/duplicate assessment (`sample_duplication_flag` values of $1-5$; see Sect. 3.3 and Table 2). The percent column indicates what percent of the original number of cores was retained after error/duplicate assessment.

| Sources | Region | n | final n | % |
|---|---|---|---|---|
| Bauer et al. (2024)[a] | Canada | 769 | 739 | 96.1 |
| Beilman et al. (2009)[a] | West Siberian Lowlands | 23 | 23 | 100 |
| Benfield et al. (2021)[a] | Sierra Nevada del Cocuy (Eastern Colombian Andes) | 22 | 20 | 90.9 |
| Cole et al. (2015) | Sarawak, Malaysian Borneo | 3 | 3 | 100 |
| Comas et al. (2015) | West Kalimantan, Indonesia | 8 | 8 | 100 |
| Crezee et al. (2022) | Central Congo Basin | 1558 | 1558 | 100 |
| Davies et al. (2021) | Southern Hudson Bay Lowlands | 1 | 1 | 100 |
| Davies et al. (2023a) | Western Hudson Bay Lowlands | 2 | 2 | 100 |
| Davies et al. (2023b) | Western Hudson Bay Lowlands | 3 | 3 | 100 |
| Gorham et al. (2012)[a] | North America | 1685 | 1478 | 87.7 |
| Hribljan et al. (2023) | Colombian, Ecuadorian, Peruvian, and Bolivian Andes | 25 | 24 | 96.0 |
| Hugelius et al. (2020)[a] | N of 23°N | 7738 | 6899 | 89.2 |
| Kelly et al. (2020) | Quistococha, Pastaza-Marañón Foreland Basin, Peru | 29 | 29 | 100 |
| Keys and Henderson (1987); Thibault (1992)[a] | New Brunswick, Canada | 20 505 | 20 505 | 100 |
| Group of A. Gallego-Sala (unpublished) | >60°N, 5°S − 5°N | 230 | 230 | 100 |
| Lawson et al. (2023) | Pastaza-Marañón Basin, Peru | 280 | 280 | 100 |
| Manitoba Dept. of Natural Resources and Northern Development[a] (unpublished) | Manitoba, Canada | 1709 | 1598 | 93.5 |
| Lamentowicz (2005) | Poland | 14 | 14 | 100 |
| Group of M. Lamentowicz (unpublished) | Poland | 249 | 249 | 100 |
| Scottish Government (2025) | Scotland, U.K. | 174 159 | 170 052 | 97.6 |
| Silvestri et al. (2019a, b) | Kubu Raya District, West Kalimantan, Indonesia | 63 | 63 | 100 |
| Treat et al. (2017, 2019)[a] | Global | 614 | 486 | 79.2 |
| Sun et al. (2023)[a] | Tibetan Plateau | 146 | 145 | 99.3 |
| Warren et al. (2012)[a] | Indonesia | 33 | 32 | 97 |
| M. Warren (unpublished) | Indonesia | 276 | 275 | 99.6 |
| Winton et al. (2025) | Colombia | 186 | 186 | 100 |

[a] indicates sources that are confirmed to be a compilation of other datasets.





## 3.2 Data Formatting

All collected data were processed into a consistent format. Source datasets were first converted to a CSV file format, if not already. Any measurements that were missing a latitude, longitude, or depth value were removed. All peat depth values were converted to centimetres and the coordinates of each measurement location were converted to the World Geodetic System 1984 (WGS84 or EPSG:4326) coordinate system, where required. When the depth measurement was presented as a range (this occurred in less than five measurements), the median of the values was determined and used within Peat-DBase. All datasets

were then added to the Peat-DBase, which is structured as a single CSV file. The columns of this synthesized database are explained in Table 2.

## 3.3 Data Processing

Data entry errors require careful handling. As described in Sect. 3.2, incomplete measurements (e.g., missing latitude or longitude values) were excluded during initial data processing. However, some complete records contained obvious errors,

such as peat measurements with coordinates placing them in the ocean. These clearly erroneous entries were flagged by setting the `error_found` column to `True` and documenting relevant details in the `investigation_notes` field (see Table 2).

Since Peat-DBase incorporates compiled datasets that are themselves compilations (see Table 1, e.g. Treat et al., 2017; Hugelius et al., 2020), additional quality control steps are necessary to handle duplicate entries. As Sarracino and Mikucka (2017) demonstrated, duplicates in modelling datasets can bias regression estimates, particularly when their distribution is

non-random – a common occurrence when the same peat depth measurements appear across multiple compilations. To address this issue, we reassessed the database for duplicates each time a new source dataset was added. Rather than removing duplicates automatically, we flagged them to allow users to retain one measurement while filtering out redundant entries as needed (see Table 2 for columns used in duplicate assessment). This approach – retaining one instance while flagging others – was among the most effective strategies for reducing bias tested by Sarracino and Mikucka (2017), outperforming alternatives such as

ignoring duplicates, removing all instances, or applying weighting schemes.

Duplicate flagging occurred in two phases. First, we identified exact duplicates – measurements with identical depth values and coordinates – and flagged all but the first instance with `sample_duplication_flag` values of 6 or 8 (depending on whether the duplication is occurring within the same source dataset or not, see Table 2) with the first instance given a value of 3. In the second phase, we deliberately reduced measurement precision to detect potential rounding by previous data sources. This

process identified measurements with matching depth values when rounded to 0.5 cm and coordinates rounded to 0.01°, then flagged them for manual assessment. These precision thresholds were determined iteratively by testing progressively coarser rounding until the number of confirmed duplicates became minimal. We note that the rounded duplicates identification step was not applied within the Scottish Government (2025) dataset because of the high precision available for its sample coordinates.

For each potential duplicate, we examined the measurements and their citation information to determine whether they orig-

inated from a common study when possible. Measurements were flagged if they came from compiled datasets containing processed or rounded versions of data already present in Peat-DBase from their original sources. For example, both Sun et al.





**Table 2.** The column headers within Peat-DBase and their meaning.

| Peat-DBase v.1 columns | Meaning |
| --- | --- |
| `original_dataset` | A citation of the publication or owner of the source dataset. |
| `original_entry_num` | Location of the measurement in the ordering of its original dataset. |
| `lat` | Latitude in decimal degrees with original number of significant figures retained. |
| `lon` | Longitude in decimal degrees with original number of significant figures retained. |
| `depth_cm` | Basal peat depth measurement in centimetres. |
| `sample_date` | Date the sample was collected[a]. |
| `site_condition` | Condition of the site where the sample was collected[a]. |
| `original_dataset_source_notes` | Any citation information provided by the `original_dataset` publication or owner. |
| `peat_measurement` | `True` if the original_dataset is a primary field study[b]. `False` if the data comes from WoSIS (see Sect. A). |
| `error_found` | `True` if possible errors (such as incorrect coordinates). Otherwise `False`. |
| `investigation_notes` | Notes on the nature of the error in `error_found`. |
| `location_is_duplicate` | `True` if there are more than one measurement in the database for the given latitude and longitude ( these measurements may or may not be duplicates of one another; see `sample_duplication_flag`.). Otherwise `False`. |
| `location_id` | An identifier unique to each location, i.e. each unique combination of latitude and longitude in the database. |
| `sample_duplication_flag` | Numerical flag: 1 - Sample obtained from a primary field study[b] confirmed in/by the original publication/owner. No further duplicate assessment performed. 2 - Sample not obtained from a confirmed primary field study[b] and not detected as a possible duplicate. 3 - Sample found to be the first instance of an exact[c] duplicate. 4 - Sample found to be the first instance of a rounded[d] duplicate. 5 - Sample found to be a redundant instance of a rounded[d] duplicate, but we have low confidence that this is a true duplicate. 6 - Sample found to be a redundant instance of an exact[c] duplicate. 7 - Sample found to be a redundant instance of a rounded[d] duplicate and we have high confidence that this is a true duplicate. 8 - Sample not obtained from a confirmed primary field study[b] and found to potentially be the duplicate of another measurement within the same source dataset. |
| `sample_id` | A group identifier unique to each group of duplicates in the database. Samples identified as duplicates are assigned the same `sample_id`. |

[a]Presently only available for the Scottish Government (2025) data. [b]Primary field study means the authors of the publication or dataset owners collected the data themselves.
[c]Exact duplicate means that the lat, lon, and depth_cm values are identical. [d]Rounded duplicate means the lat and lon values are identical when rounded to 2 decimal places and the depth_cm values are within 0.5 cm.





(2023) and Treat et al. (2017) incorporate data from Zhao et al. (2014), converting the original arc-minute coordinates to decimal degrees. Since Treat et al. (2017) retained fewer significant figures, their duplicate entries were flagged with values of `7` for the `sample_duplication_flag`. However, incomplete citation practices in some datasets prevented definitive
conclusions. Measurements confirmed to originate directly from field studies were always retained, and we applied conservative criteria throughout—keeping measurements unless clear evidence of duplication existed (e.g. exact same coordinates with high precision and exact same depth). If we had low confidence in our assessment of a rounded duplicate, we applied a `sample_duplication_flag` value of `5` indicating the potential for the measurement to be a duplicate but also conveying the uncertainty in that assessment.

As mentioned above, we retain all measurements in Peat-DBase, regardless of their `sample_duplication_flag` and `error_found` values. We do this to allow future refinement of these assessments, enhance traceability back to their original data sources, and to ensure our assessments can be audited as needed. When using Peat-DBase we suggest the following filtering then be applied: 1) select only `error_found` values of `False`, and 2) select `sample_duplication_flag` values of `1`, `2`, `3`, `4`, and `5`. We note that some applications of Peat-DBase may require stringent removal of duplicates, in that
case, `sample_duplication_flag` values of `5` should be removed as well.

## 4 Results and Discussion: Overview and Characteristics of Peat-DBase

### 4.1 General overview

Following harmonization and quality control procedures, Peat-DBase version 1.0 comprises 204 902 measurements from 29 peat-focused sources (Table 1). When combined with non-peat study data from WoSIS, the database contains 299 517 mea-
surements (Figure 2). The peat study data spans 54.933°S to 82.217°N, a latitudinal range that remains unchanged when incorporating non-peat study data.

Figure 3 reveals that peat study data is heavily concentrated in the northern extratropics, with particularly high sample densities in New Brunswick, Canada and Scotland. This concentration reflects the inclusion of two large source datasets (Keys and Henderson, 1987; Thibault, 1992; Scottish Government, 2025). Users of Peat-DBase should be aware of this uneven spatial
distribution, as it may introduce regional bias depending on the intended application. For global-scale analyses, sub-sampling these data-rich regions may be necessary to prevent New Brunswick and Scotland from exerting disproportionate influence on results.

Figure 2a shows that most peat depth measurements exceed 30 cm – a common threshold for peatland classification (Loisel et al., 2017). Among the peat-focused studies, 5 831 measurements (2.8%) report a zero cm peat depth, primarily from sampling
schemes that measured across transects to assess peatland presence, such as those by Crezee et al. (2022) and Keys and Henderson (1987).

When incorporating filtered WoSIS data, zero-depth measurements increase to 100 446 (Figure 2b). Figure 2b also highlights notable data gaps in desert regions like the Sahara and in certain countries such as Paraguay. These gaps exist in the original WoSIS database before any filtering described in Sect. A3 and are further discussed by Batjes et al. (2020b).



**Figure 2.** Distribution of data points in Peat-DBase version 1.0. (a) Only the data points originating from peat-focused sources (primary field studies or compilations) without errors or duplicates, i.e. `sample_duplication_flag` values of 1 − 5; see Sect. 3.3 and Table 2, are shown. (b) As in subplot a, but including those filtered from WoSIS. Note, the colour bar has a log scale with a colour break at 30 cm to visually delineate a common peat depth threshold for classification as a peatland (Loisel et al., 2017).

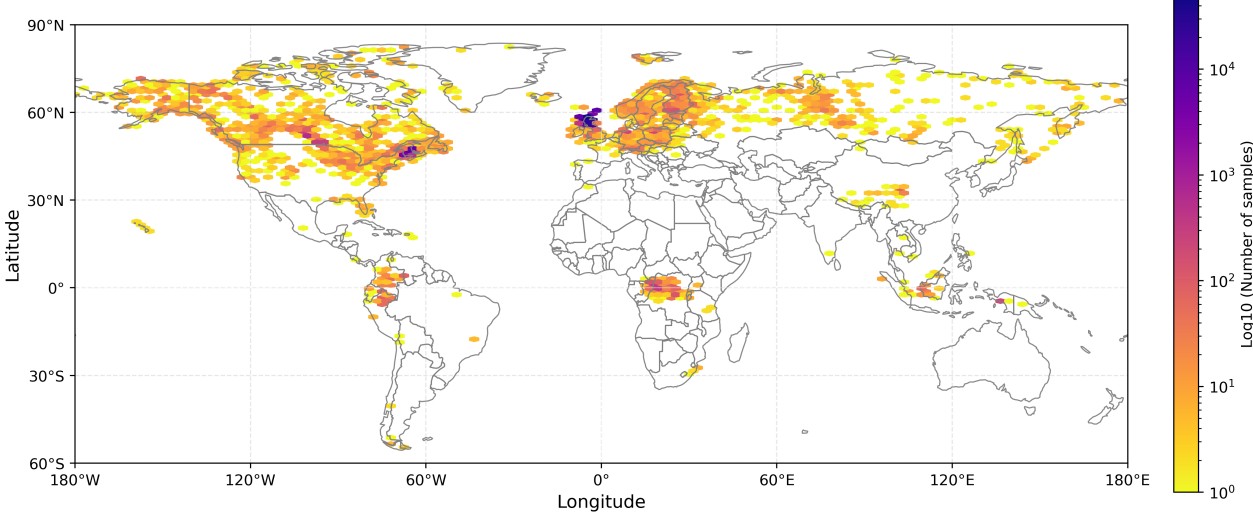

**Figure 3.** Map of Peat-DBase sample density for peat measurements as in Figure 2a. Bin sizes are 3° in longitude. Note the log scale.

## 4.2 Spatial and Depth Distribution of Data

Peat-DBase represents most major global peatland areas. The distribution of peat depth measurements (Figure 2a and Figure 3) broadly aligns with peatland fractional coverage shown in published global products: PEATMAP (Xu et al., 2018) and Peat-ML (Melton et al., 2022) and in the tropics-only CIFOR (Gumbricht et al., 2017). PEATMAP synthesizes the most detailed regional peatland maps available prior to 2018, while Peat-ML uses machine learning to predict global peatland coverage by training on regional peatland maps and environmental variables (Melton et al., 2022). Conversely, CIFOR is an expert-informed system predicting peatland area by integrating moisture supply, soil saturation, and geomorphological characteristics (Gumbricht et al., 2017).

Peat-DBase includes measurements from most major peatland complexes identified in both PEATMAP and Peat-ML, including those in North America, Eurasia, South America, the Congo Basin, and the Malay Archipelago (Figure 2a). However, coverage gaps exist in the Amazon Basin, Indonesia, and Papua New Guinea, where PEATMAP and CIFOR indicate extensive peatland presence (Xu et al., 2018; Gumbricht et al., 2017). Peat-ML similarly shows greater peatland coverage in these regions and additionally in Eastern Russia. While paleoecological evidence supports substantial peatland presence in Eastern Russia (Yu et al., 2010, Figure 1), we have not identified additional readily available peat depth datasets for this region. African peatlands are particularly poorly documented across the scientific literature (Gallego-Sala et al. in review 2025) and Peat-DBase also appears to also poorly represent the full extent of peatlands on the continent.

While Peat-DBase captures a significant proportion of our current knowledge on peat depth, it also reflects inherent biases arising from field research constraints. The database shows notably sparse coverage in low-latitude regions (Figures 2, 3, 4), consistent with the historically less extensive mapping of tropical peatlands (Zinck, 2011; Ruwaimana et al., 2020). Although





this tropical data gap shapes the current distribution within Peat-DBase, ongoing research efforts (e.g., Peat-ML2 - discussed
in Sect. 4.3 and Winton et al. (2025)) are attempting to address these knowledge gaps, and future studies will be incorporated
into subsequent database versions as they become available.

The distribution of measurements in Peat-DBase reflect the prominence of peatlands in boreal, temperate, and tropical
regions (Xu et al., 2018; Melton et al., 2022; Joosten and Clarke, 2002; Koster and Favier, 2005). Looking beyond the high
sample densities in New Brunswick and Scotland, Figure 4 and Figure 3 shows peat measurements concentrated in high
latitudes (particularly 40°N–50°N) and near the equator. Additional clusters appear at intermediate latitudes, such as around
35°N, corresponding to peatland complexes in Florida and the Tibetan Plateau (Figure 2a). The deepest recorded measurement
within Peat-DBase (3 527 cm) occurs in the Tibetan Plateau (Sun et al., 2023).

The deepest peat deposits typically occur in regions that either escaped glaciation during the Last Glacial Maximum or were
among the first to become ice-free during deglaciation, allowing extended accumulation periods (Treat et al., 2019; Ruwaimana
et al., 2020; Gowan et al., 2021). Favourable topographic settings, such as flat floodplains and narrow river basins, also facilitate
deep peat formation (Figure 2, 4, and see Figure 1 in Treat et al. (2019)). However, peat depth does not correlate linearly with
age, as peatlands undergo variable accumulation rates influenced by changing climatic and hydrologic conditions, including
periods of enhanced growth, stagnation, or erosion (Ruwaimana et al., 2020; Blaauw and Christen, 2005; van Bellen et al.,
2011).

The depth distribution of non-zero measurements in Peat-DBase, which comprise 66.5% of the database, can be fit with
a Weibull Minimum distribution (red line in Figure 5). When WoSIS data are excluded, the database composition shifts to
non-zero peat depths constituting 97.2% of measurements, with a mean depth of 144 cm.

Most peatland field studies lack spatial scaling considerations, resulting in non-random, clustered distributions around re-
search sites of interest (Hugelius et al., 2020; Meyer and Pebesma, 2022). This spatial clustering prevents direct comparisons
between peat-to-non-peat ratios in Peat-DBase and actual peatland coverage.

Figure 5 reveals decreasing data availability with increasing peat depth. While this pattern may reflect natural peat devel-
opment, sampling bias likely contributes. Deeper coring presents logistical challenges: standard equipment typically handles
depths up to several hundred centimetres, while depths exceeding 1000 cm require specialized strategies and equipment (Bansal
et al., 2023; Shotyk and Noernberg, 2020).

Our prioritization of large datasets for Peat-DBase version 1.0 may have excluded smaller studies or single-core inves-
tigations that reached substantial depths. Counterbalancing that bias, researchers often target presumed peatland centres –
typically the deepest areas – for paleo-reconstructions, accumulation rate estimates, or carbon stock assessments. These stud-
ies frequently collect limited cores (Hugelius et al., 2013; Hribljan et al., 2016; Loisel et al., 2017), potentially skewing the
database's depth distribution toward deeper measurements.

Limited core sampling can affect the representativeness of peat depth distributions in Peat-DBase. In peatlands developed
over flat mineral basins with uniform surfaces, relatively few measurements may adequately capture depth variability. How-
ever, peatlands formed in complex topography or with variable surface gradients require more extensive sampling to accurately
represent depth distributions (Hugelius et al., 2020; Loisel et al., 2017). van Bellen et al. (2011) illustrate this variability using



**Figure 4.** Distribution of measurements by depth (a), by depth and latitude (b), and by latitude (c). WoSIS data are excluded from all panels. Additionally, for this plot, we removed data points noted as drained or modified. At present, this information is only available for the observations derived from the NatureScot dataset (Scottish Government, 2025).





**Figure 5.** Peat depth distribution within Peat-DBase version 1.0 on a log scale, 94 615 measurements come from WoSIS (see Sect. A). The red line indicates the Weibull Minimum distribution (calculated via SciPy; Virtanen et al., 2020) of the non-zero cm depth measurements within Peat-DBase. Note the log scale is applied to both the x and y axes. As this plot focuses on depth distributions, we removed data points noted as drained or modified. At present, this information is only available for the observations derived from the NatureScot dataset (Scottish Government, 2025).





probing, coring, and ground-penetrating radar to reveal surface altitude variations of two to eight metres within individual
peatlands. Their findings show that maximum depths do not necessarily occur at geographic centres due to underlying basin
topography. To address these sampling challenges, national or regional inventories, such as those from NatureScot (Scottish
Government, 2025) and the Government of New Brunswick (Keys and Henderson, 1987; Thibault, 1992), and peatland map-
ping initiatives employ systematic transects and comprehensive sampling strategies designed to capture diverse peat formations
(Hugelius et al., 2020; Crezee et al., 2022; Silvestri et al., 2019a; Parry et al., 2012). These broader sampling approaches can
help mitigate depth representation bias within Peat-DBase.

### 4.3   Database Limitations and Future Work

Several uncertainties and limitations should be considered when using Peat-DBase. The database does not reflect current peat-
land status, as we included historical measurements regardless of present-day conditions. Consequently, some measurements
may originate from peatlands that have since been degraded or destroyed by land-use change or climate impacts (Joosten
and Clarke, 2002; Koster and Favier, 2005; Ratnayake, 2020; Silvestri et al., 2019a). Future versions will incorporate data on
current peatland status to address this limitation. This enhancement is particularly important for applications utilizing Earth
observation (EO) data, such as high-resolution land cover datasets, where temporal mismatches can create inconsistencies.
Land surface changes occurring after peat core extraction –such as conversion to urban or industrial uses – can create conflict-
ing information between recorded peat depths and current land use, potentially compromising data utility for contemporary
analyses.

The literature lacks consensus on peat definition, introducing classification uncertainties into Peat-DBase (Lourenco et al.,
2022; Gumbricht et al., 2017; Page et al., 2011; Zinck, 2011; Magnan et al., 2024). Organic matter thresholds vary widely:
Silvestri et al. (2019a) define peat as containing at least 30% organic matter, while Cole et al. (2015) and Crezee et al. (2022)
require at least 65%. Currently, Peat-DBase does not track these varying definitional criteria; however, future versions will
incorporate this information to ensure the influence of classification decisions on data interpretation remains traceable.

Measurement accuracy in Peat-DBase reflects the technological constraints of each study period. Many measurements from
the 1950s through 1990s predate GPS technology or used early, less accurate receivers (Treat et al., 2017; Hugelius et al., 2020;
Sun et al., 2023; Keys and Henderson, 1987). Historical data storage methods also pose challenges for the incorporation of
known, existing data, with some datasets existing only in formats difficult to access or process (Thibault, 1992).

Field measurement techniques introduce additional uncertainties. Metal probes may encounter false resistance from buried
wood fragments or interbedded mineral layers from ash (Hribljan et al., 2016), fluvium (Lähteenoja et al., 2012), or colluvium
before reaching the true peat base (Parry et al., 2014). Coring can compress peat layers, leading to underestimated depths
(Shotyk and Noernberg, 2020).

WoSIS shares similar sampling uncertainties with the peat study components of Peat-DBase. Batjes et al. (2020b) docu-
ment variability in geographic coordinate precision and laboratory measurements within WoSIS, along with corresponding
uncertainty metrics. However, since Peat-DBase focuses on peat presence rather than precise mineral soil characterization,
we did not incorporate these WoSIS uncertainty measures when identifying non-peat locations. Nevertheless, variability in





geographic coordinate precision affects Peat-DBase data from peat-focused sources and will be investigated in future versions
to provide positional uncertainty estimates. These estimates are particularly important when using peat measurement data with
high-resolution Earth observation (EO) data, as positional uncertainty can result in measurements being erroneously placed in
oceans, on mountain tops, or other environments unsuitable for peat formation.

Our duplicate assessment (Sect. 3.3) may introduce subjective bias, as it relies on our interpretation of reasonable evidence
of coordinate rounding. Furthermore, we did not test all possible decimal placements due to rounding, potentially missing both
subtle and extreme rounding instances.

While some regions lack available peat depth data (Sect. 4.2), other areas have data not yet incorporated into Peat-DBase.
Known examples include peat depth measurements from the Amazon Basin near the Madre de Dios River (Householder et al.,
2012) and various Indonesian sites (Anda et al., 2021). These datasets were excluded because they lacked readily accessible
point-based formats. Additionally, our prioritization of large datasets may have overlooked publications with single or few
measurements. Future database versions will aim to incorporate these missing sources and additional datasets as they become
available. For example, the Can-Peat project (https://uwaterloo.ca/can-peat/) is currently developing a database of >100 000
Canadian peat depth measurements by digitizing and collating data from both published literature and mandated environmental
impact assessments (pers. comm. A. Dalton, June 2025). We are aware of another large dataset (>18 000 measurements) focused
on Alberta, Canada (pers. comm. K. Bona, July 2025) that will also be incorporated as it becomes available.

## 5 Conclusions

Peat-DBase version 1.0 represents the most comprehensive global-scale compilation of peat depth data currently available.
With over 200 000 measurements from peat-focused studies alone, it substantially expands when incorporating non-peat soil
data. The database's spatial distribution largely aligns with established peatland coverage maps, though notable gaps remain
in under-sampled regions. While sampling bias influences the depth distribution, the database serves dual purposes: providing
essential data for global-scale analyses and highlighting geographic and depth-range gaps that warrant future research attention.
Peat-DBase is presently under active development to both expand the number of measurements and improve its relevance and
accuracy.

## 6 Data availability

Peat-DBase version 1.0 is stored in a CSV file located here https://doi.org/10.5281/zenodo.15530645 (Skye et al., 2025).

Peat-DBase is under active development under a Google grant, *Peat-ML2: A new global benchmark for global peatland*
*carbon inventories*, awarded to JRM and RSW. We encourage interested data contributors to fill out a short survey found at
https://forms.gle/WzbUpPQFZMK3Yyme6 to join the project or contact JRM, RSW, or LS.





## Appendix A: Non-Peat Study Data

Non-peat study data were added to Peat-DBase to provide representation of non-peat regions. Given that non-peat areas were not the primary focus of Peat-DBase, detailed soil profiling was not prioritized. Rather, the goals in acquiring non-peat data
were broad land coverage and confirmation of mineral soil presence. Measurements from these areas were assigned a peat depth of zero cm.

### A1  WoSIS

WoSIS maintains a harmonized and quality-controlled database of global soil profiles for digital soil mapping purposes. Existing soil data are submitted by owners for consideration in WoSIS, where they are stored, assessed, and standardized through
the WoSIS workflow. Iterations of the fully quality-assessed and standardized database are released periodically as snapshots. The September 2019 snapshot was acquired for use in Peat-DBase. This snapshot is documented in Batjes et al. (2020b) and the data are available through Batjes et al. (2019).

### A2  WoSIS Data Formatting

The non-peat study data were processed to the same format as the peat-focused study data. The WoSIS database is stored
across several TSV files. Only the `wosis_201909_profiles.tsv` and `wosis_201909_layers_chemical.tsv` files were required for subsequent processing steps; these were converted to CSV file format. These files contain soil profile coordinates in WGS84 format and the chemical properties of soil profiles, respectively (Batjes et al., 2020b). This information was used to determine which cores represented mineral soil profiles and therefore non-peat cores.

### A3  WoSIS Data Processing

The acquired soil profile data were filtered to include only mineral soil profiles. Our study broadly follows the peatland definition suggested by Lourenco et al. (2022), which specifies an area with a minimum of 5% organic carbon content to a minimum depth of 10 cm. Here we treat soils with less than 5% organic carbon content as mineral soils for the purpose of establishing peat-free locations. The WoSIS dataset often contains organic carbon content measurements in g/kg for multiple layers within a soil profile, although not all soil profiles have associated organic carbon content data (Batjes and Van Oostrum, 2023; Batjes
et al., 2020b). Therefore, the first processing step was to exclude all soil profiles with no organic carbon content measurements, since their qualification as mineral soil based on our 5% organic carbon threshold could not be readily determined otherwise. Next, all organic carbon content measurements were converted from gC/kg of soil mass to percent mass. Any profiles containing a layer with organic carbon content greater than or equal to 5% were then excluded from the dataset. The coordinates of all remaining soil profiles were collected as a new dataset, and these locations were assigned a peat depth of zero cm.

The data derived from WoSIS were not subject to duplicate assessment within this study. Duplicate assessment was deemed not necessary within the dataset itself, as assessment and exclusion processes were applied prior to the WoSIS snapshot release by the dataset authors (Batjes et al., 2020b). Regardless, we note there remains some entries from the WoSIS database within





Peat-DBase that have the same `location_id` and with a `depth_cm` of 0. Users may need to filter those entries depending on their need as they may also be considered duplicates (e.g. when training machine-learning models). Duplicate assessment

was additionally not conducted between the derived mineral soil cores and the peat study database, as we assumed significant duplication between mineral soil profiles and peat study data would be unlikely.

*Author contributions.* Conceptualization, Funding acquisition, and Project administration: JRM; Methodology: JS, JRM, CG, AG-S, RSW, MG, LS; Investigation and Data Curation: JS, LS; Formal analysis: JS, LS, JRM; Resources (unpublished data): AG-S, JCB, GI, EBB, CK, FK, MM, VM, MMV, ML, DW, SC, JaL, JoL, MW, LF; Resources: RSW, LS, ML, LESC, MAD, EAL, JS, YW, JRM, CG; Software: JS,

LS; Validation: JS, LS, JRM; Visualization: JS, JRM; Writing – original draft: JS, JRM, CG; Writing – review & editing: All; Supervision: JRM, CG

*Competing interests.* The authors declare no competing interests

*Acknowledgements.* JS was supported by the Federal Student Work Experience Program through Environment and Climate Change Canada, and a University of Victoria Faculty of Graduate Studies Fellowship. LS was supported by a Google grant, "Peat-ML2: A new global bench-

mark for global peatland carbon inventories", awarded to JRM and RSW. AGS was funded by the European Research Council (ERC) under the European Union's Horizon 2020 research and innovation programme (grant agreement No 865403) and by the UK Natural Environment Research Council under agreement number NE/S001166/1. ML, DW, JaL and JoL were funded by the National Science Centre, Poland, grant no 2021/41/B/ST10/00060. The Polish fieldwork was done within the scope of the project "Protection of Valuable Ecosystems of Tuchola Forest" funded by the European Economic Area Financial Mechanism 2014-2021 within the framework of the Environment, Energy

and Climate Change Programme MF EEA 2014-2021 "Implementation of Ecosystem Management Plans". We thank Emily Prystupa for supplying data on behalf of the Department of Natural Resources for the Government of New Brunswick; Kelly Bona and Kara Webster for helping with access to the Bauer et al. (2024) dataset; Esther Lévesque for her assistance collecting samples in Nunavut. For unpublished data contributed by the group of AG-S, we acknowledge the community of Mittimatalik (NU), the Centre d'études nordiques and Parks Canada (permit # SIR-2021-39579) for access to the site as well as the logistical support of the Polar Continental Shelf Program (Natural

Resources Canada) and the financial support of the Natural Sciences and Engineering Research Council of Canada (Discovery program). YW acknowledges a PhD scholarship via the QUEX Institute (The University of Queensland and The University of Exeter). LC acknowledges NERC, the NERC Radiocarbon Dating Facility (Radiocarbon Analysis Allocation Number 1565.0411) and SUERC Dating Lab for their financial support. AI chat engines were used to assist coding tasks relating to Peat-DBase visualization and copy editing of the manuscript text written by the authors.



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
