# Peer review of "Peat-DBase v.1: A Compiled Database of Global Peat Depth Measurements"

_Earth System Science Data, 2025_

## Author Response (AR1)

Dear Professor Loisel,

Please find below the comment from your group in black font. Our reply is provided in blue font. We appreciate your group both reading our manuscript and providing a community comment. We also appreciate being notified that the very deep Tibetan point was in error (switched elevation and peat depth), which we have now rectified.

—

We are a group of graduate students who have read your study with interest. For context, we discuss 1-3 scientific papers that relate to peatland dynamics weekly.

The study reads very well and is adequately referenced. We understood the context, objectives, and the flow of information was always relevant to addressing the study's goals.

Thank you for your comment. We are glad to hear our study was chosen by your group for its journal club - and that you brought back your comments here allowing us to improve our manuscript based on them.

One issue that was raised regards the definition of 'peat'. Depending on the threshold used, the peatland extent could be more than 3%. Also, we appreciate that all peat depth data are included, making the 'peat / no-peat cutoff' at the discretion of the user, but it could be good to explicitly state this fact in the document. In other words, maybe add a sentence to the effect that all peat depths have been included.

We have changed this line (line 85 in original MS; italics show the new text):
"Peat study data were accepted into Peat-DBase provided the measurements were taken down to the basal depth indicated by mineral soil or bedrock; beyond this requirement, any coring or sampling method was allowed that was able to accurately determine the basal depth. *We included peat depths (from peat-focused studies) starting from 0 cm (indicating mineral soils) and deeper, as we wish to allow Peat-DBase users to use their own peat depth definitions for their particular use case.*"

Based on Figure 2, it looks like the vast majority of the shallowest peats (less than 30 cm; in red on panel A) are in the Congo Basin. Are we sure that these individual points referred to 'peat depths' in the original paper (Crezee et al. 2022), or did the original authors include lots of points aimed at showing areas without peat? (we believe the latter is correct). We also noticed a few of those red points in the Colombian lowlands. Lastly, the reds vs. greens may not be colorblind friendly.

Following our approach of taking in all observations from peat-focused studies we do end up with some 0 cm depth observations. In particular, the Crezee et al. (2022) paper has many peat

observations that appear to correspond to villages that we assume they drove through during the course of their field work, e.g.

[Figure]

We feel these points are useful to include as they constrain where peat is absent in these peat-rich regions.

We did run all figures through a colour-blind simulator and the figure seems to work well in the simulated renderings:
Original:

[Figure]

Simulated for different forms of colour blindness:

[Figure]

[Figure]

[Figure]

Data availability: we like the ease of downloading a single CSV file that contains the database and the opportunity to share new data points via the Google form.

We are glad to hear, we also find the CSV simple to work with.

Discussion: we appreciate the focus on peat depth alone, without trying to correlate with other predictive factors. We anticipate that future work will be based on such types of analysis, given the inclusion of the (non-peat) WoSIS data.
Thank you. This approach was partly enforced by the journal specifications but also made sense for the purpose of this paper in our view.

—------------

Dear Professor Frolking,

Thank you for your review. Please find below your comments in black font. Our reply is provided in blue font.

**Steve Frolking:**

This manuscript presents a new public data set of global peatland depths, compiled from published literature, national or regional datasets, and some unpublished data.  This is an important new data set for global carbon cycle assessments and earth system modeling activities.  ESSD is an appropriate journal outlet. The paper is very well written.

I have no major questions/issues with the manuscript. All of my comments and questions below are minor ones.

Thank you for your positive assessment of our manuscript.

Line 60: 'often contain errors' seems a bit harsh for a ubiquitous problem.  maybe just 'can contain errors'?

True, there were a few of the compilations that did not have errors (shown in Table 1), changed to 'can contain errors'.

Line 61: 'outdated measurements': I don't know what this means.

Our comment here refers to the fact that a peatland can be measured as an intact peatland and then later experience disturbance either natural or anthropogenic in which case the peat depth may have then changed dramatically. We have reworded this for clarity (new text in italics), "Current peatland compilations assembled for individual studies (e.g. Gorham et al., 20212; Treat et al., 2019; Hugelius et al., 2020) can contain errors, duplicates, outdated observations *(whereby a peatland has experienced natural or anthropogenic disturbance since measurement)*, among other issues."

Line 68: Not sure if this is necessary, but it might be worth noting that most land-surface models need peat depth data for initiation as they do not simulate the multiple millennia needed to accumulate the peat, and the peat stocks are not necessarily in equilibrium. They cannot be 'spun up' to an equilibrium state.

Yes, this was an important motivating factor in initiating our study. The text as is does mention the requirement for data for model initialization, i.e. "First, advancing peatland representation in land surface models (Wu et al., 2016; Bechtold et al., 2019s; Apers et al., 2022) requires accurate, extensive data for initialization and evaluation, especially as these models are integrated into Earth system models."

In Fig. 1, step 1 is to 'confirm basal depths'. To me this implies confirm that the values are correct (how?) but maybe it means 'confirm reported peat depths are basal'?  Also, step 3 includes 'remove unnecessary information', which makes sense.  In the best of all possible data worlds, it would be easy to align Peat-DBase data with other data sets (e.g., one of basal ages, or one of age-depth profiles).  The burden of creating this best-of-all-possible-worlds shouldn't fall on you, but it seems to me that it must be something you have thought about. Would any of this 'unnecessary data' be useful in this regard?  Would that introduce uncertainties that you don't want to (and shouldn't be expected to) have to manage?  Could this be discussed in the 'future work' section, perhaps identified as 'future work for the community to move the field forward', not specifically as future work for Peat-DBase and this manuscript's authors.

"Confirm basal depths" has now been expanded for clarity to be "Confirm peat depths correspond to basal depth" and updated in Figure 1:

[Figure]

Step 1: Confirm peat depths correspond to basal depth

Step 2: Format to Peat-DBase CSV conventions

Step 3: Convert to Peat-DBase units, remove unnecessary information, and flag erroneous entries

Step 4: Add dataset to Peat-DBase and prepare for duplicate assessment

Step 5: Perform duplicate assessment

Indeed, as we have continued development of Peat-DBase, we have begun bringing in this 'other' information as we feel it could be valuable especially as the dataset could be used for estimating important quantities like global peatland carbon stores. At present, all new sites include extensive information beyond peat depth (including allowing non-basal depth measurements) like organic carbon and bulk density measurements and we are working to add in information for sites already in the database. We have added some information about the expansion to Peat-DBase that we are presently undertaking (Line 245 in original MS, new text in italics):"Future versions will incorporate data on current peatland status to address this limitation. … Peat-DBase will also be expanded to include organic carbon content and bulk density data for the soil cores, as well as characteristic information about the sampling site such as hydrological status, soil water pH, vegetation present, etc. "

Line 125: are some of the peat depth data in the ocean from Treat et al. 2019, which included coastal shelf peatland data from the last glacial?  If so, these are not in error, just not relevant for Peat-Dbase. Also, there is not universal agreement on where the land ocean boundary is, and it certainly depends on spatial resolution.

Thank you for this comment. Of the 614 points for Treat et al. (2017; 2019), 5 were flagged as being in the water. These were located tens of kilometres from the nearest land point. We have added a postscript to Table 1 explaining: "Five of the cores in Treat et al. (2019) were excluded due to being located in the present-day ocean. These sites were not in error, but were collected to characterize peatlands across the last glacial cycle when the sites were subaerial."

Table 2. 'depth_cm': is that reported to any particular significant figure?  nearest integer cm?

We recorded the values as presented in the original sources.We have changed that line in Table 2 to, "Basal peat depth measurement in centimetres *with precision as recorded in the original source*".

Table 2 and elsewhere: for the sample duplication flag, one data sample is the 'first instance'. How is this selected/determined: earliest published, first that you acquired, first that you entered, ...?  It likely doesn't matter in terms of the database, but it would be good to explain what you mean by 'first instance' as priority in time is an important currency in academic publication.

The determination of the 'first instance' is based on several factors. We have added the following description to the text discussing Table 2 for clarity:
*"Detected duplicates were sorted by precision and source. For rounded duplicates, the point with the greatest decimal precision was designated as the first instance. For exact duplicates, points were sorted alphabetically by dataset name (e.g., Gorham et al. (2012) before Hugelius et al. (2020)). When datasets were added incrementally, previous duplicate assessments were retained, and newly added data were compared only against existing first instances and unduplicated records. Consequently, first instance designation reflects both the sorting criteria and the temporal sequence of data acquisition, which followed the order datasets were obtained (except datasets with publication restrictions, which were added last in their acquisition order)."*

Fig. 3 caption: Bin sizes are 3° in longitude; how many in latitude?

This comment has prompted us to more accurately record the bins sizes. The caption now reads: "Bin sizes are *approximately 2.88° in longitude and 2.5° in latitude.*"

Line 213: 'enhanced growth, stagnation, or erosion' -- I think that 'loss' (i.e., from 'excess' decomposition, not erosion) should be added to this list.

Indeed, now reads: "However, peat depth does not correlate linearly with age, as peatlands undergo variable accumulation rates influenced by changing climatic and hydrologic conditions, including periods of enhanced growth, *loss*, stagnation, or erosion".

Fig. 4 caption: it would be useful to add binning sizes (50 cm?, 1° latitude?)

Yes, we have now added to the caption, "Data has been binned by 2° in latitude and 50 cm of depth."

Fig. 5: The color scale is not too helpful here, at least for me, as the x-axis is number of measurements.  I think it would be helpful to label the bar at the top 'zero depth', since it doesn't really work on the vertical log scale (I initially mis-interpreted this as the sum of all other data, but, of course, it didn't add up).  In the caption: I don't understand why a focus on depth distributions leads you to remove data points labeled as drained or modified. Particularly since

you cannot make this assessment for all data. Nonetheless, if you maintain this, I suggest adding an 'n = NN' in parentheses for number excluded.

We have revised the figure to make it more useful. We removed all non-peat cores (WoSIS and zero depth cores from peat-focused studies). We then adjusted the colour scale to emphasize the distribution better. We have also kept out the 'drained/modified' cores as their depths could be highly modified from 'natural' cores. We have modified the caption "...  As this plot focuses on depth distributions, we removed data points noted as drained or modified (n=110 924). At present, this information is only available for the observations derived from the NatureScot dataset \citep{scotland_peatdepth2025}. Figure A1 is a version of this plot with those measurements included." We include this revised figure in the main text:

[Figure]

and this new figure (with drained/modified samples) added to the appendix:

[Figure]

More generally on the drainage/disturbance question: you say in the captions to Figs. 4 & 5 that drainage information is only available for the NatureScot dataset. How is that noted in the Peat-Dbase (it is not mentioned in Table 2)? Others may also want to make the exclusion that you did.

In Table 2, the column 'site_condition' contains the information needed to exclude the drained/disturbed sites.. At present the information is only available for NatureScot but we are working to expand it on other sites in future versions of Peat-DBase.

-Steve Frolking

—--------

Dear Reviewer #2,

Thank you for your review. We post your comments in black font and our reply in blue.

**Anonymous Referee #2:**

Skye et al., present a timely examination of global peat depths, based on the synthesis of 25 regional/global prior syntheses. The paper presents the motivation, analysis, results, and discussion well and I have no major issues with the current manuscript. I have several minor comments, mostly focused on how to deal with the definition of "peat" and the measurement of peat depth in a "meta-synthesis" such as this paper.

Thank you for your positive assessment of the work.

In definitions of peat that include lower OM%, it is often more difficult to characterize when "basal peat" is reached as opposed to mineral soils. I agree with the author's decision to include everything described as peat by the primary source, but I think that should be explicitly stated early on in Sec. 3.1.

We have edited the start of Section 3.1 to be, '... *To select studies we relied on the authors' identification of what constitutes peatlands or peat soils and their determination of the basal peat depth*.'

Additionally, while I appreciate the inclusion of non-peat mineral soils into the database to highlight regions where peat is not found, I think one of the recent maps of peat area should be included in the main text figures. This helps to visually highlight peat-rich regions that are currently under-sampled better than simply listing a few examples in the text.

We have now added Peat-ML (Melton et al. 2022) to Figure 3 to allow easier comparison.

[Figure]

Other Minor Comments:

Line 85 – "Peat study data…" This sentence is somewhat awkwardly phrased.

We have restructured it to 'Peat study data were accepted into Peat-DBase provided the measurements were taken down to the basal depth indicated by mineral soil or bedrock\footnote{Future versions of Peat-DBase will allow peat data that does not reach the basal depth through the use of appropriate flags indicating that fact.}. Any coring or sampling method was allowed that could accurately determine this basal depth.'

Line 91 – While you discuss limitations of the dataset elsewhere, I would rephrase this sentence "until they meet a non-peat layer and cannot go any further" as probes can still penetrate non-peat sediments without meeting refusal. This is especially true of very humified, lower

organic % peats which often grade into organic-rich mineral soils (i.e. humic silt sediments) found in more temperate regions.

This is a good point. We have added, 'Probing, involves the use of metal poles which are inserted into the peat until they meet a non-peat layer and cannot go any further \citep[e.g. Oakfield probes;][]{Magnan2024-dz,Householder2012-sh,Crezee2022-sx}. *We note probing has higher uncertainty for estimating peat depths as there is no visual confirmation of the interface between the peat and the underlying substrate.*'

Line 100 – Where there any systematic methods used for searching the literature for peat depths?

There was no structured approach to the search beyond checking common databases such as Google Scholar and UVic libraries for literature with applicable key words. Results of these searches were broadly prioritised based on whether the titles or abstracts suggested the collection/assemblage of a significant amount of peat cores (e.g. Treat et al. (2019), Hugelius et al. (2020), Crezee et al. (2020), etc.). Literature with potentially large amounts of data were prioritised as it helped to reduce data processing time and, in the instances where the data was assembled from other sources (e.g.. Treat et al. (2019), Hugelius et al. (2020)), they already contained the data from other literature that appeared in searches.

Figure 2a – I'm not sure why the mineral soils are plotted for the Congo region but not elsewhere? Are those wetland soils that have a mineral substrate? i.e. hydric soils?

The mineral soil cores in other peat study data papers are all plotted, however plotting small grid points like this on a global scale leads to over plotting and obscures the data points. We experimented with smaller points but it makes it harder to see the data in other regions. These points in the Crezee et al. dataset appear to be cores taken in villages or at least were recorded as non-peat points, e.g.

[Figure]

Figure 3 – You discuss comparisons with PEATMAP in the main text (line 183). Could you add the peat map as a background to your figure on Peat-DBase data density? This would be the most "major" revision I would recommend.

We agree with this suggestion being helpful to a reader and have added the Peat-ML map (Melton et al. 2022) to Figure 3 as shown earlier.

Line 200 – While I agree that the source data is biased to the northern hemisphere, the overall total land area coverage of tropical peatlands is also lower than boreal peatland coverage.

We agree, tropical peatlands have smaller extent than those in the boreal. However, even accounting for that, we are biased to the high-latitudes as our tropical peatland cores are heavily clustered leaving many areas of the tropics missing data.

Figure 5 – The color bar scale should be reduced, as currently there is not much variation in the color for each bin.

Thank you, based on this comment and those of Steve Frolking we have revised Figure 5 to be:

[Figure]

251 – This is a minor comment, but I might put this paragraph first as I think it's the most pressing issue to resolve in future versions of the dataset.

We agree that uncertainty around definitions is a pressing issue, but prefer our present wording. Given how much peatland area is now under anthropogenic pressure (e.g. Fluet-Choinard et al. (2023)), we feel that understanding whether the core is from a pristine or degraded/modified peatland is important as these changes could drastically change the peat depth.

285 – If one of the major goals was to highlight under sampled regions, then I would definitely add one of the recent peat-map areas to the map in Figure 3 as there is currently not a great way for the reader to judge visually which peat-rich areas are under sampled.

We have addressed this now with our revised Figure 3 shown above.

References cited:

Crezee, B., Dargie, G. C., Ewango, C. E. N., Mitchard, E. T. A., Emba B., O., Kanyama T., J., Bola, P., Ndjango, J.-B. N., Girkin, N. T., Bocko, Y. E., Ifo, S. A., Hubau, W., Seidensticker, D., Batumike, R., Imani, G., Cuní-Sanchez, A., Kiahtipes, C. A., Lebamba, J., Wotzka, H.-P., Bean, H., Baker, T. R., Baird, A. J., Boom, A., Morris, P. J., Page, S. E., Lawson, I. T., and Lewis, S. L.: Mapping peat thickness and carbon stocks of the central Congo Basin using field data, Nature Geoscience, 15, 639–644, https://doi.org/10.1038/s41561-022-00966-7, 2022.

Fluet-Chouinard, E., Stocker, B. D., Zhang, Z., Malhotra, A., Melton, J. R., Poulter, B., Kaplan, J. O., Goldewijk, K. K., Siebert, S., Minayeva, T., Hugelius, G., Joosten, H., Barthelmes, A., Prigent, C., Aires, F., Hoyt, A. M., Davidson, N., Finlayson, C. M., Lehner, B., Jackson, R. B.,

and McIntyre, P. B.: Extensive global wetland loss over the past three centuries, Nature, 614, 281–286, https://doi.org/10.1038/s41586-022-05572-6, 2023.

Gorham, E., Lehman, C., Dyke, A., Clymo, D., and Janssens, J.: Long-term carbon sequestration in North American peatlands, Quaternary Science Reviews, 58, 77–82, https://doi.org/10.1016/j.quascirev.2012.09.018, 2012.

Hugelius, G., Loisel, J., Chadburn, S., Jackson, R. B., Jones, M., MacDonald, G., Marushchak, M., Olefeldt, D., Packalen, M., Siewert, M. B., Treat, C., Turetsky, M., Voigt, C., and Yu, Z.: Large stocks of peatland carbon and nitrogen are vulnerable to permafrost thaw, Proceedings of the National Academy of Sciences, 117, 20 438–20 446, https://doi.org/10.1073/pnas.1916387117, 2020.

Melton, J. R., Chan, E., Millard, K., Fortier, M., Winton, R. S., Martín-López, J. M., Cadillo-Quiroz, H., Kidd, D., and Verchot, L. V.: A map of global peatland extent created using machine learning (Peat-ML), Geosci. Model Dev., 15, 4709–4738, https://doi.org/10.5194/gmd-15-4709-2022, 2022.

Treat, C. C., Kleinen, T., Broothaerts, N., Dalton, A. S., Dommain, R., Douglas, T. A., Drexler, J. Z., Finkelstein, S. A., Grosse, G., Hope, G., Hutchings, J., Jones, M. C., Kuhry, P., Lacourse, T., Lähteenoja, O., Loisel, J., Notebaert, B., Payne, R. J., Peteet, D. M., Sannel, A. B. K., Stelling, J. M., Strauss, J., Swindles, G. T., Talbot, J., Tarnocai, C., Verstraeten, G., Williams, C. J., Xia, Z., Yu, Z., Väliranta, M., Hättestrand, M., Alexanderson, H., and Brovkin, V.: Widespread global peatland establishment and persistence over the last 130,000 y, Proceedings of the National Academy of Sciences, 116, 4822–4827, https://doi.org/10.1073/pnas.1813305116, 2019.